# Estimation of Morphological Features and Essential Oil Content of Basils (*Ocimum basilicum* L.) Grown under Different Conditions

**DOI:** 10.3390/plants11141896

**Published:** 2022-07-21

**Authors:** Danguolė Juškevičienė, Audrius Radzevičius, Pranas Viškelis, Nijolė Maročkienė, Rasa Karklelienė

**Affiliations:** 1Department of Vegetable Breeding and Technology, Lithuanian Research Centre for Agriculture and Forestry, Kauno 30, Kaunas Distr., 54333 Babtai, Lithuania; audrius.radzevicius@lammc.lt (A.R.); nijole.marockiene@lammc.lt (N.M.); rasa.karkleliene@lammc.lt (R.K.); 2Laboratory of Biochemistry and Technology, Lithuanian Research Centre for Agriculture and Forestry, Kauno 30, Kaunas Distr., 54333 Babtai, Lithuania; pranas.viskelis@lammc.lt

**Keywords:** cultivar, fresh herb, morphological traits, *Ocimum basilicum* L., productivity

## Abstract

This study was carried out in the experimental field and in unheated greenhouses of the Lithuanian Research Centre for Agriculture and Forestry Institute of Horticulture. The investigation aimed to evaluate the morphological features, amount of essential oil, and productivity of 10 basil (*Ocimum basilicum* L.) cultivars grown under different growing conditions. Studied cultivars were different according to morphological parameters, productivity, and the accumulation of essential oil. Most of the investigated parameters were influenced by growing conditions. Basil plants of cultivars Sweet Genovese and Toscano were determined to be the highest, and their height reached up to 70.5 cm. Cultivar Sweet Genovese was the most productive; its fresh herb mass per plant reached up to 0.71 kg under growing in the open field and greenhouse. Higher air temperature and constant watering of the soil in the greenhouse increased the accumulation of essential oils of basil. The biochemical analyses showed that the essential oil amount of 31% was higher when basils were grown in a greenhouse compared to an open field.

## 1. Introduction

Basil (*Ocimum basilicum* L.) is grown in many countries of the world, and it is widely used for its medical and aromatic properties [1,2]. These properties are associated with the presence of essential oil in basil. Many studies and conclusions are presented about the composition and peculiarities of essential oil accumulation in basils as well as in other spices and plants with antioxidant properties [3,4,5,6,7,8,9,10,11,12,13,14]. Chang et al. [15] report the content and amounts of essential oils depending on the age of basil plants. The growth, yield, plant quality, and nutrition of basil under soilless agricultural systems are studied [16,17,18,19]. It is proved that the aeroponic system records the highest values of basil roots, shoots and leaves, and essential oil content compared to hydroponic and peat moss slabs [18]. Walters and Currey [17] determine the influence of different hydroponic systems on the efficiency of basil production. One of the latest experiments confirms the effect of genotype and season on the content of ascorbic acid and nitrate levels in plants [19]. Barickman et al. [20] present and the dependence of morphological processes on the air temperature and CO^2^ content. 

Basil is mostly cultivated in the countries with the warmer climate in Tropical America, Africa, Asia, and Southern EU [15,21,22,23]. It is an annual herb native to India and other parts of Asia [23]. Previous studies have shown that basil is susceptible to growth temperature. Meteorological and growth conditions are one of the most important factors that affect the yield quality and essential oils accumulation of basils [2,5,13]. Temperature below 10 °C impairs their growth [24,25]. Recent studies by Barickman et al. [20] show that decreasing the growth temperature to 20/12 °C in day/night is the major factor positively determining for basil’s morphological features.

Basil productivity and the level of essential oil are also influenced by genetic factors [2,16,25]. Numerous basil cultivars and forms are currently cultivated in different countries of the world and in Europe. They differ according to the plant size and habit, color, shape, and size of leaves and flowers as well as the content and composition of essential oil, and other biologically active substances [5,9,22,26]. According to the Lithuanian climatic condition, basil is an annual plant, and two forms var. *latifolia* and var. *rubra* are cultivated [14,27]. A wide diversity of new foreign forms and cultivars, their morphological and chemical variability changing environmental factors, and the importance of growing conditions optimization require further studies about the cultivation of basils.

Having in mind that the productivity of basil is influenced by a complex of factors, including the climatic conditions, agrotechnological measurements, and genotypic effect the hypothetical statements of the study were the observation of different genotypes’ ability to adapt to Lithuanian climatic conditions. Additionally, it is actual to find the most optimal growing method of basil cultivars originated from the warmer regions for the fresh mass increase and the accumulation of essential oil.

This study aimed to investigate the morphological features, amount of essential oil, and productivity of different basil cultivars growing in the open field and the unheated greenhouse and to evaluate the influence of meteorological and different growing conditions.

## 2. Results

### 2.1. Morphological Features and Productivity of Basil

The plant’s height varied from 44.0 to 77.6 cm in the greenhouse and from 37.2 to 63.4 cm under growing in the open field (Table 1). Three phenotypes, cvs. Sweet Genovese, Fino Verde, and Toscano, were significantly the highest among the cultivars in both growing conditions. The average height of these cultivars reached 76.4 cm in the greenhouse and 61.5 cm in the open field. Cultivar Cinnamon formed the highest plant foliage in the greenhouse; its perimeter reached up to 76.7 cm, while the height reached up to 64.8 cm. A similar size tendency of plant foliage was observed in the cv. Toscano in these conditions. The perimeter of foliage reached up to 74.1 cm and the height up to 77.6 cm, respectively. Observations of leaf morphological parameters showed that cv. Toscano plants formed the widest and the longest leaves under field conditions. The average width was determined to be up to 5.3 cm and length up to 11.2 cm, respectively. Cultivars Classic Italiano and Toscano formed the widest leaves in the greenhouse, up to 5.2 and 5.9 cm, respectively. Additionally, cv. Toscano formed the longest leaves, reaching 14.8 cm.

Morphological parameters of basil were influenced by meteorological conditions. The majority of studied cultivars, grown under field conditions, formed higher and wider foliage in the wetter season of 2017. The average foliage height reached 53.7 cm and perimeter reached 51.2 cm in 2017, which was 10% and 8% higher compared to 2018 (Table 2). Two cultivars, Toscano and Cinnamon, formed the highest and the widest foliage in 2017. Moreover, the plants of cv. Toscano were taller and wider in 2018. The average foliage perimeter of basils grown in greenhouse was similar and reached 62.4 cm in 2017 and 63.3 cm in 2018, while the height of plants in 2017 was 11% higher. 

Results of the basil productivity showed differences between investigated cultivars in fresh mass and net mass. Fresh mass of basil varied from 0.12 to 0.62 kg per plant in the open field and from 0.13 to 0.79 kg per plant in the greenhouse (Table 3).

The highest fresh mass was produced of cv. Sweet Genovese. The productivity of cv. Fino Verde reached up to 0.55 kg per plant under field conditions and up to 0.64 kg per plant in the greenhouse, respectively. These results are like those of other researchers presenting that cv. Fino Verde’s productivity reached up to 0.71 kg per plant [26]. Two cultivars, Toscano and Fino Verde, distinguished with the highest net mass on the field conditions. It reached 83.2 and 84.4%, respectively. Three cultivars, Toscano, Classic Italiano, and Fino Verde, showed significantly similar results in net mass compared to other cultivars in the greenhouse. The net mass was from 83.5 to 83.5%. The rest of the cultivars were similar by the net mass and varied from 71.0 to 81.1% in the field and from 76.8 to 80.5% in the greenhouse.

Our results showed that the influence of meteorological conditions to the productivity of different basil cultivars was low. The average fresh mass of plants grown under field conditions reached 0.42 kg per plant and was the same in 2017 and in 2018. The net mass was higher by 4% in 2018 (Table 4). Meteorological conditions in 2018 was more favorable for basil grown in the greenhouse. The fresh mass was 17% higher compared to 2017, while the net mass was similar. Cultivar Sweet Genovese was the most productive in both years of the study.

### 2.2. Accumulation of Essential Oil

Accumulation of essential oil is one of the most desirable and important features of basil. Analysis showed that basil accumulated the essential oil up to 0.12% grown under field conditions and up to 0.20% in the greenhouse during the two years of investigation (Table 3). Basil cv. Holy distinguished by the highest possibility for the essential oil accumulation in the greenhouse. Under these conditions, three cultivars, Siam Queen, Cinnamon, and Compact, showed the same essential oil accumulation. Their amount of essential oil reached 0.18%. The same cultivars, Siam Queen, Cinnamon, and Holy, showed the highest possibility for the essential oil accumulation under field conditions, where the amount of essential oil reached 0.12%. 

### 2.3. Influence of Genetic Origin and Growing Conditions

Summarizing the results of the investigated basil cultivars while plants were grown under both conditions showed the diversity among genotypes. Two cultivars, Sweet Genovese and Toscano, were distinguished with the highest average plant height of up to 68.7 and 70.5 cm, while cv. Cinnamon formed the widest average foliage (Table 5). The foliage perimeter of cv. Cinnamon reached up to 70.8 cm. The results of productivity showed that, on average, 0.48 kg of fresh herb per plant were obtained. Significantly, the most productive were two cultivars Classic Italiano and Sweet Genovese, when the amount of fresh herb reached 0.66 and 0.71 kg per plant, respectively. The productivity of cv. Holy was the lowest (0.13 kg per plant) among tested cultivars, but it accumulated the highest amount of essential oil. Its amount of essential oil reached up to 0.16%. A similar amount of essential oil reaching 0.14–0.15% was obtained from cvs. Siam Queen, Cinnamon, and Compact.

Estimation of the influence of growing conditions showed that the significantly higher effects of plant and foliage morphometric parameters, the productivity of the fresh herb, and the accumulation of essential oil were observed in the experiments when basils were grown in the unheated greenhouse (Table 6).

The principal coordinate analysis (PCoA) displayed the diversity of all investigated cultivars for morphological features, productivity, and essential oil formation. According to the PcoA results of the plant height and perimeter, it is possible to classify the investigated samples into two groups. Four cultivars, Sweet Genovese, Toscano, Siam Queen, and Fino Verde, showed the possibility to form a higher plant and wider foliage. They dispersed in one group in the PCo scatter plot area with the highest positive value (Figure 1). The rest of the cultivars were in the second group on the front side of the PCo scatter plot not far away from each other.

The PCoA of different basil cultivars according to the productivity and net mass on both growing conditions showed that investigated samples dispersed on two sides of the scatter plot. On the one side of the scatter plot, 60% of the studied cultivars were located (Figure 2). The productivity of these cultivars reached from 0.36 to 0.71 kg per plant and the net mass reached from 79.3 to 84.1%, respectively. Three cultivars, Cinnamon, Compact, and Purple Opal, were located on the separate side of the scatter plot. Their productivity was lower and reached from 0.34 to 0.50 kg per plant and the net mass from 73.9 to 80.6%. Cultivar Holy had formed the lowest productivity and a high net mass located in the individual point of PCoA scatter plot.

The PCoA of essential oil accumulation presented a wide location of cultivars (Figure 3). Three cultivars, Siam Queen, Cinnamon, and Compact, were located in one area of the PCoA scatter plot. The average amount of essential oil in these cultivars reached 0.14–0.15%. Cultivar Holy, accumulating the highest amount (0.16%) of oil, scattered on the same side of the PCoA area.

## 3. Discussions

The influence of meteorological conditions and different ways of growing to the basil growth and accumulation of essential oils were observed in our study. All cultivars showed a good adaptivity to our environmental conditions. They were different in productivity, morphological features, and essential oils amount. Various researchers confirm a similar dependence of such variations on the genetic origin and climatic conditions [1,2,15,24,27,28,29].

The meteorological conditions, especially amount of precipitation, were different during the study years and influenced the growth of basil. A higher and wider foliage of basils was determined in the wetter season of 2017, compared to 2018. The increase in moisture in the soil could have influenced better root development and optimal plant nutrition conditions. Meteorological conditions had less influence for plants’ morphometric parameters in the greenhouse. However, air temperature and a 10% higher level of solar radiation in 2018 compared to 2017 could be more favorable for the formation of basil fresh mass in greenhouse. Cultivar Sweet Genovese, originated in the USA, was the most productive in both years of the study.

Our results showed that the growth of basil was influenced by the growing conditions. The average basil height reached up to 51.4 cm and foliage diameter reached up to 51.4 cm when plants were grown in open fields. The average height of plants up to 61.3 cm and foliage perimeter up to 62.8 cm were obtained in the greenhouse. Similar results according to such parameters were reported by other authors. The basil plant heights reach up to 57 cm and foliage perimeters up to 61 cm when plants are grown under field conditions [28].

Basil fresh mass reached up to 0.42 kg per plant when plants were grown under open field conditions and up to 0.52 kg per plant in the greenhouse, respectively. A higher temperature, constant watering, and lower temperature deviations during the basil growing in the greenhouse had a more positive effect on the plant development and growth. 

A similar influence of growing conditions was observed in the assessment of essential oils accumulation. Our results showed a better accumulation of essential oil when plants were grown under greenhouse conditions. Cultivar Compact accumulated a twofold higher amount of essential oil in the greenhouse (0.18%) compared with the open field. It is presented that basil growing is more effective in warmer regions [24,30]. The oil amount ranges up to 0.8% in winter and up 0.5% in summer while temperatures reach 35–40 °C in Pakistan [16]. A decrease in oil amount in summer might be explained by the high temperatures and partial volatilization of some constituents of oil. The essential oil yield reaches up to 0.58% when the air temperature during the experiment is fixed between 20 and 27 °C [31]. A higher output of essential oils is recorded in summer and a lower output in winter in the Southern hemisphere [32]. Researchers report the accumulation of essential oil depending on the genotype. The yield of the essential oils obtained from aerial parts of Iranian cultivars is 0.2and 0.5% [29]. Cultivars Siam Queen and Cinnamon show the highest ability of essential oil accumulation among the tested twelve cultivars in Serbia [33]. Our results showed that amount of essential oils of mentioned cultivars reached up to 0.18%. In Lithuania, the yield of essential oils of cv. Sweet Genovese reached 0.10%, but Švecová and Neugebauerová [28] present up to 0.18% in this cultivar. 

## 4. Material and Methods

### 4.1. Plant Material 

The study was carried out in the experimental field and unheated greenhouse at the Lithuanian Research Centre for Agriculture and Forestry Institute of Horticulture (LRCAF IH) in 2017–2018. Experiments were repeated for two years. Ten basil cultivars were investigated (Table 7).

### 4.2. Plant Cultivation

Seedlings of plants were produced from seeds in the glasshouse. Seeds were sown into containers filled with peat and sand (3:1) (pH 5.5–6) in the second ten days of April and placed in a heated glasshouse with the temperature ~14 ± 5/23 ± 5 °C night/day. The seedlings were transplanted to the unheated greenhouse and the field 30–35 days after seeds germination in the first ten days of June. The distance between the greenhouse and field was 10 m and the soil was similar in places of trial. The soil was characterized by a neutral pH, relatively low humus levels, and medium plant-available phosphorus and potassium contents. Every year before planting, 50 gm^−2^ of complete mineral fertilizers (YaraMila™) (N-12, P_2_O_5_-11, K_2_O-18) was applied. During the experiment in the greenhouse, plants were watered every 3–4 days and the greenhouse was aired. Watering under field conditions was not used. No diseases or pests were observed, hence plant protection was not used.

Plants were grown on a flat surface. The distance between plants was 30 cm and 70 cm inter-rows. The investigation was carried out in four replications every year. Seven plants were grown in one plot and repeated 4 times, for a total of twenty-eight plants of each cultivar. Basil was harvested in the second ten days of August when plants began the period of flowering. This occurred after 80–90 days of sowing. The harvest was carried out by cutting plants 1.5–2.0 cm above the soil. After the cutting, plants were weighed and the average mass of fresh basil herb per plant (kg) was determined. Manual separation of leaves, buds, and inflorescences was made for the evaluation of fresh net mass (℅).

### 4.3. Morphological Measurements

In the first ten days of August, the diameter and height of plants, width, and length of leaves were measured for assessment of morphological features. Each cultivar was represented by twelve plants when three plants from four replications were measured. For the assessment of leaves’ width and length, ten of the best-developed leaves were measured. 

### 4.4. Essential Oil Quantification

The essential oil of basil was determined at the Laboratory of Biochemistry and Technology (LRCAF Babtai, Kaunas distr., Lithuania). Buds, inflorescences, and leaves without stems were used for the estimation of essential oil accumulation. Parts of the plant were dried at 35 °C in a Universal Oven ULE 500 (Memmert GmbH+Co. KG, Schwabach, Germany) as air-dry plants. The essential oil was prepared from dried powdered plant material (100 g) by the Clevenger-type distillation (USP XXII, NFXVII, 1990) [34] on a laboratory scale, and the amount of essential oil was calculated based on the air-dry mass of the plant materials.

### 4.5. Meteorological Conditions

The weather data were collected at the Babtai Agrometeorological station, located 100 m from the field using the METOS^®^sm prognostication system The mean air temperature in June was 16.1 °C and reached 15.3 °C and 16.9 °C in 2017 and 2018, respectively (Figure 4). The average temperature in July varied from 16.7 °C (2017) to 20.5 °C (2018), while the temperature during the last vegetation period of basil in August reached up to 19.3 °C in 2018 and up to 17.6 °C in 2017. Precipitation amount during the basil vegetation was 57% higher in 2017 compared to 2018. June was rather dry in 2018 when the amount of precipitation was 17.8 mm, and it reached up to 40.4 mm in 2017, respectively. A similar tendency was seen in July and August. The amount of precipitation was two times higher in July 2017 compared to 2018, when in August the amount up to 58.8 mm reached in 2017 and up to 21.8 mm in 2018. The solar radiation during the basil vegetation (June, July, August) reached 554,82 W m^−2^ in 2017 and 613,42 W m^−2^ in 2018, respectively.

The thermometer scale showed the temperature range of 15–32/10–15 °C in day/night in the unheated greenhouse during plant vegetation.

### 4.6. Data and Statistical Analysis

The data of morphological features, productivity, and essential oil of basil were calculated and presented as averages for both years. They were statistically processed by the analysis of variance (ANOVA). Significant differences between the experimental treatments were determined using Duncan’s multiple range test at the 5% probability level (*p* < 0.05), and the analysis of principal coordinates SPSS 11.5 software package (SPSS Inc., Chicago, IL, USA). The results are presented in the PCoA scatterplot, which indicates distinct differences between productivity, morphological features, and amount of essential oil in basils of different cultivars.

## 5. Conclusions

All cultivars adapted well to Lithuanian climatic conditions. Differences in the productivity, morphometric parameters, and essential oils accumulation of basil cultivars from different countries were influenced by growing conditions and genetic origin. More massive foliage of plants was determined in the study year, when the amount of precipitation was higher.

Estimation of the influence of growing conditions has shown that higher air temperature and constant watering in the soil in the greenhouse more effectively influenced morphometric parameters and productivity of basils. Basils grown in a greenhouse accumulated more essential oils compared to those grown in the open field.

Basil cultivars Sweet Genovese and Toscano distinguished with the highest foliage, and cultivar Cinnamon formed the widest foliage. Cultivars Classic Italiano and Sweet Genovese were the most productive. The highest ability for essential oil accumulation was shown by cultivar Holy.

## Figures and Tables

**Figure 1 plants-11-01896-f001:**
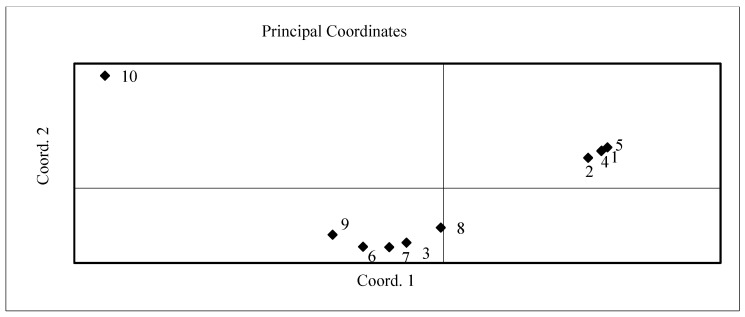
Scatter plot of different basil cultivars according to the results of principled coordinate analysis (PCoA) of plant height and foliage perimeter estimation in 2017–2018. Basil cultivars: 1—Sweet Genovese; 2—Toscano; 3—Classic Italiano; 4—Siam Queen; 5—Fino Verde; 6—Cinnamon; 7—Compact; 8—Purple Opal; 9—Rosie; 10—Holly.

**Figure 2 plants-11-01896-f002:**
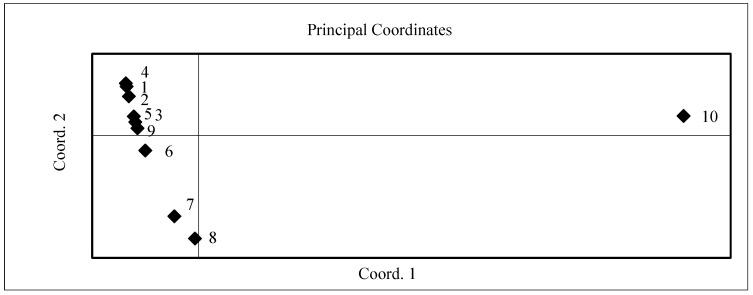
Scatter plot of different basil cultivars according to the results of principled coordinate analysis (PCoA) of productivity and net mass in 2017–2018. Basil cultivars: 1—Sweet Genovese; 2—Toscano; 3—Classic Italiano; 4—Siam Queen; 5—Fino Verde; 6—Cinnamon; 7—Compact; 8—Purple Opal; 9—Rosie; 10—Holly.

**Figure 3 plants-11-01896-f003:**
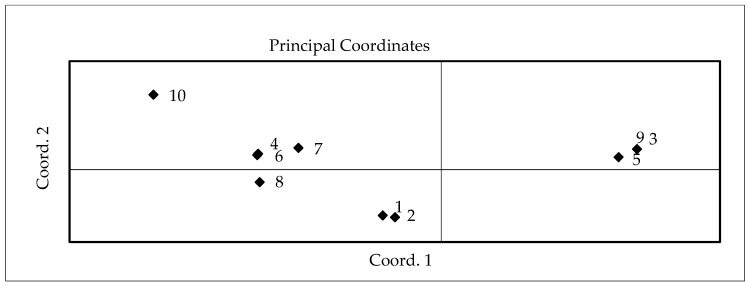
Scatter plot of different basil cultivars according to the results of principled coordinate analysis (PCoA) of the essential oil accumulation in 2017–2018. Basil cultivars: 1—Sweet Genovese; 2—Toscano; 3—Classic Italiano; 4—Siam Queen; 5—Fino Verde; 6—Cinnamon; 7—Compact; 8—Purple Opaal; 9—Rosie; 10—Holly.

**Figure 4 plants-11-01896-f004:**
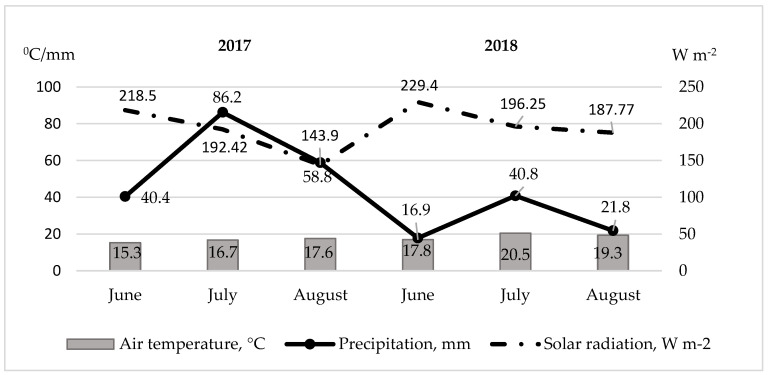
Meteorological conditions.

**Table 1 plants-11-01896-t001:** Morphological parameters of basils grown in field and greenhouse.

Cultivar	Plant Height, cm		Foliage Perimeter, cm		Leaf Width, cm		Leaf Length, cm	
	Field	Greenhouse	Field	Greenhouse	Field	Greenhouse	Field	Greenhouse
Sweet Genovese	60.8 a	76.5 a	59.3 b	72.6 b	2.7 d	4.0 b	6.8 cd	9.9 c
Toscano	63.4 a	77.6 a	59.4 b	74.1 ab	5.3 a	5.9 a	11.2 a	14.8 a
Classic Italiano	59.0 b	71.7 ab	57.7 b	71.8 b	4.5 b	5.2 a	8.8 b	10.7 b
Siam Queen	42.1 de	46.2 c	48.0 ab	54.9 cd	2.6 d	2.8 c	7.2 cd	8.1 c
Fino Verde	60.3 a	75.2 a	59.8 ab	72.3 b	2.1 de	2.6 c	5.8 d	7.5 d
Cinnamon	59.4 a	71.4 ab	64.8 a	76.7 a	3.9 c	4.1 ab	8.5 b	9.2 c
Compact	37.2 e	45.7 ac	40.8 d	52.8 d	1.9 e	2.3 c	5.4 d	7.0 d
Purple Opal	42.0 de	45.1 c	43.6 c	53.6 d	4.5 b	4.5 ab	8.9 b	11.0 b
Rosie	48.4 c	58.3 bc	42.2 cd	57.7 c	3.5 c	3.8 b	8.3 b	9.3 bc
Holy	37.9 e	44.0 c	32.9 e	41.5 e	2.8 d	3.2 b	6.7 cd	6.6 d
Average	51.4	61.3	50.8	62.8	3.3	3.8	7.7	9.4

Means followed by the same letter did not differ significantly within the column at *p* = 0.05 (Duncan’s multiple range test). (*n* = 4 × 2).

**Table 2 plants-11-01896-t002:** Morphological parameters of basil foliage in 2017 and 2018.

Cultivar	2017				2018			
	Plant Height, cm		Foliage Perimeter, cm		Plant Height, cm		Foliage Perimeter, cm	
	Field	Greenhouse	Field	Greenhouse	Field	Greenhouse	Field	Greenhouse
Sweet Genovese	63.4 ab	81.1 a	59.8 b	73.4 a	58.1 b	71.9 a	57.8 b	71.7 b
Toscano	65.2 a	83.1 a	59.6 b	73.1 a	63.6 a	72.1 a	58.2 ab	75.3 a
Classic Italiano	61.1 b	83.4 a	58.4 b	71.3 b	57.2 b	61.9 b	56.5 b	72.3 b
Siam Queen	45.0 cd	46.4 c	48.0 c	54.7 c	39.1 d	46.0 d	47.2 c	55.2 cd
Fino Verde	62.7 ab	79.3 b	60.0 b	71.9 b	57.9 b	71.1 a	58.7 ab	72.7 ab
Cinnamon	66.7 a	72.4 c	65.7 a	76.0 a	58.1 b	70.3 a	63.4 a	77.4 a
Compact	40.1 d	50.0 e	40.5 d	52.4 cd	34.3 e	41.4 e	41.2 d	55.2 cd
Purple Opal	45.4 cd	44.1 f	44.2 cd	53.6 c	38.6 d	46.3 d	42.5 d	53.5 d
Rosie	50.4 c	63.6 d	42.4 d	57.0 c	46.4 c	52.9 c	41.0 d	58.4 c
Holy	35.9 ef	48.1 ef	33.4 e	40.9 e	39.8 d	39.9 ef	32.0 e	41.4 e
Average	53.7	65.2	51.2	62.4	49.3	57.4	47.5	63.3

Means followed by the same letter did not differ significantly within the column at *p* = 0.05 (Duncan’s multiple range test). (*n* = 4).

**Table 3 plants-11-01896-t003:** Productivity and amount of essential oil of basils grown in field and greenhouse.

Cultivar	Fresh Mass, kg *		Net Mass, %		Essential Oil, %	
	Field	Greenhouse	Field	Greenhouse	Field	Greenhouse
Sweet Genovese	0.62 a	0.79 a	80.3 bc	78.3 bc	0.10 b	0.10 de
Toscano	0.46 d	0.63 cd	83.2 ab	83.5 ab	0.08 c	0.11 d
Classic Italiano	0.57 bc	0.74 ab	81.1 b	83.5 ab	0.09 bc	0.11 d
Siam Queen	0.32 fg	0.39 f	80.6 bc	80.0 b	0.12 a	0.18 b
Fino Verde	0.55 c	0.64 cd	84.4 a	83.8 a	0.02 e	0.16 c
Cinnamon	0.42 e	0.49 e	81.1 b	80.2 b	0.12 a	0.18 b
Compact	0.43 e	0.58 de	76.8 bc	80.1 b	0.09 bc	0.18 b
Purple Opaal	0.31 fg	0.38 f	71.0 bc	76.8 bc	0.08 c	0.04 f
Rosie	0.44 d	0.47 ef	68.2 c	73.8 bc	0.06 d	0.06 e
Holy	0.12 g	0.13 g	79.3 bc	80.5 b	0.12 a	0.20 a
Average	0.42	0.52	78.6	80.1	0.09	0.13

Means followed by the same letter did not differ significantly within the column at *p* = 0.05 (Duncan’s multiple range test); * Mass of fresh basil herb per plant, kg. (*n* = 4 × 2).

**Table 4 plants-11-01896-t004:** Productivity of basils grown in 2017 and 2018.

Cultivar	2017				2018			
	Fresh Mass, kg		Net Mass, %		Fresh Mass *, kg		Net Mass, %	
	Field	Greenhouse	Field	Greenhouse	Field	Greenhouse	Field	Greenhouse
Sweet Genovese	0.60 a	0.69 a	80.1 a	82.1 a	0.64 a	0.88 a	80.4 b	74.5 b
Toscano	0.49 b	0.54 b	77.0 a	80.9 a	0.42 c	0.71 b	90.5 a	86.7 a
Classic Italiano	0.58 a	0.72 a	79.2 a	81.1 a	0.55 ab	0.75 b	83.0 ab	85.9 a
Siam Queen	0.36 cd	0.40 d	81.8 a	82.8 a	0.27 d	0.37 e	79.3 b	77.2 b
Fino Verde	0.53 ab	0.60 b	78.3 a	80.9 a	0.56 ab	0.67 bc	90.5 a	86.7 a
Cinnamon	0.40 c	0.45 c	78.4 a	83.7 a	0.43 c	0.53 d	83.5 ab	76.7 b
Compact	0.41 c	0.52 bc	76.8 ab	78.6 a	0.45 bc	0.64 bc	76.7 b	81.5 a
Purple Opal	0.34 d	0.39 d	71.7 b	76.6 ab	0.28 d	0.36 e	70.3 bc	76.9 b
Rosie	0.39 c	0.47 c	74.9 b	74.0 b	0.49 b	0.46 de	61.5 d	73.5 bc
Holy	0.14 e	0.15 e	73.5 b	77.1 a	0.09 e	0.11 f	85.0 a	83.9 a
Average	0.42	0.49	77.2	79.8	0.42	0.55	80.1	80.4

Means followed by the same letter did not differ significantly within the column at *p* = 0.05 (Duncan’s multiple range test); * Mass of fresh basil herb per plant, kg. (*n* = 4).

**Table 5 plants-11-01896-t005:** Morphological futures, fresh mass, and amount of essential oil of different basil cultivars *.

Cultivar	Plant Height, cm	Foliage Perimeter, cm	Fresh Mass, kg **	Essential Oil, %
Sweet Genovese	68.7 ab	66.0 b	0.71 a	0.10 b
Toscano	70.5 a	66.8 b	0.55 cd	0.10 b
Classic Italiano	65.4 c	64.8 bc	0.66 ab	0.10 b
Siam Queen	44.2 e	51.5 d	0.36 f	0.15 a
Fino Verde	67.8 b	66.1 b	0.60 c	0.09 b
Cinnamon	65.4 c	70.8 a	0.46 e	0.15 a
Compact	41.5 f	44.8 f	0.51 d	0.14 ab
Purple Opaal	43.6 e	48.6 e	0.35 f	0.06 c
Rosie	53.4 d	48.5 e	0.46 e	0.06 c
Holy	41.0 f	37.2 g	0.13 g	0.16 a
Average	56.4	56.8	0.48	0.11

Means followed by the same letter do not differ significantly within the column at *p* = 0.05 (Duncan’s multiple range test); * Average value of cultivar grown in field and greenhouse; ** Mass of fresh basil herb (kg) per plant (*n* = 4 × 2 × 2).

**Table 6 plants-11-01896-t006:** The influence of growing conditions on the morphological futures, productivity, and amount of essential oil of basils *.

Growing Conditions	Plant Height, cm	Foliage, Perimeter, cm	Fresh Mass, kg **	Amount of Essential Oils, %
Greenhouse	61.3 a	62.8 a	0.52 a	0.13 a
Field	51.4 b	50.8 b	0.42 b	0.09 b
Average	56.4	56.8	0.47	0.11

Means followed by the same letter do not differ significantly within the column at *p* = 0.05 (Duncan’s multiple range test); * Average value of all tested cultivars; ** Mass of fresh basil herb (kg) per plant.

**Table 7 plants-11-01896-t007:** Representation of cultivars.

No.	Name	Breeding Company
1.	Sweet Genovese	Johnny’s Selected Seeds, USA, Maine
2.	Toscano	Franchi Sementi, Italy, Bergamo
3.	Classic Italiano	Franchi Sementi, Italy, Bergamo
4.	Siam Queen	Johnny’s Selected Seeds, USA, Maine
5.	Fino Verde	Johnny’s Selected Seeds, USA, Mine
6.	Cinnamon	Johnny’s Selected Seeds, USA, Maine
7.	Compact	Semo Seed, Czech Republic, Smrzice
8.	Purple Opal	Semo Seed, Czech Republic, Smrzice
9.	Rosie	Terranova Seeds, Australia, Smithfield NSW
10.	Holy	Johnny’s Selected Seeds, USA, Maine

## Data Availability

Not applicable.

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
