# Peer review of "Estimation of Morphological Features and Essential Oil Content of Basils (Ocimum basilicum L.) Grown under Different Conditions"

_plants, 2022, doi:10.3390/plants11141896_

Round 1
Reviewer 1 Report
The manuscript was considerably improved, but it is still not consistent.
Abstract: The discussion was based on fact that had not been measured, soil moisture. Not correct.
Introduction: improved, but what was the hypothesis? For example, that the cultivars originated from hot/cold regions would be more productive in fresh mass/oils?
M&M:
1. Statistical analyses still not clear. Which was software, package? How did you calculate means when data were pooled?
2. Why two years that were very different meteorologically, and/or two growing conditions were pooled?
3. Improve the Meteorological data presentation and focus on diurnal/nocturnal temperatures, day length, precipitation distribution.
Results:
1. Mass not weight,
2. Plant perimeter/projection, not diameter.
3. The n for mean values is missing.
Discussion: I tried to rewrite as it could sound, impersonal, direct. You worked on summer growth and try to discuss those data and reasons for essential oil production (protection, acclimatization…)
Conclusions: Must respond to hypothesis that is missing?
Detailed observations you can find in the pdf version.

Author Response
Dear Reviewer,
thank you very for the comments and suggestions for the improving of our manuscript. Response to your comments is provided in the attached file. Please see the attachment.
Sincerely, Authors

Reviewer 2 Report
Please consider the following recommendations:
in lines 72-73: “Cultivar Cinnamon formed the highest plant foliage in the greenhouse its diameter reached up to 76.7 cm.” It should be rephrased because there are 2 morphological parameters that it is speaking about and the according values should be specified for both.
in line 115 ’Summarized´ should be replaced with „Summarizing” or „Analyzing”
in line 105 “Biochemical analysis” – what kind of biochemical analysis does the study present?
in Table 3. the * from title does not appear besides „Essential oil, %”
in lines 205-206 „A decrease in oil amount in summer might be explained by the high temperatures and partial evaporation of some constituents of oil can be expected.„ – it should be rephrased.
in line 318 “basil” not “fasil”
Correlations between morphometric parameters and essential oils accumulation of basil cultivars would be of interest.
If growing conditions influence the productivity, morphometric parameters, and essential oils accumulation of basil cultivars they should be presented more accurate and the obtained results should be analyzed by taking into consideration the growing conditions of each experimental year.
Author Response
Dear Reviewer,
Thank you for the review of our manuscript.
The response to your comments is provided in the attached file. Please see the attachment.
Sincerely yours, Authors

Round 2
Reviewer 2 Report
I agree with the revised form of the manuscript.
This manuscript is a resubmission of an earlier submission. The following is a list of the peer review reports and author responses from that submission.
Round 1
Reviewer 1 Report
Dear authors,
please find my comments in attach.

Reviewer 2 Report
Abstract: You need here some kind of conclusion. For example, why occurred this difference in EO amount. Also, you do not need to abbreviate anything not used lately in the abstract.
Introduction: Not sufficient, extremely generic, no idea, no hypothesis.
M&M:
1. Impossible to dry fresh mass for 24 hours on 35 °C.
2. Statistical analyses not clear. You used one or two way ANOVA, which method? Ptrincipal coordinate analyses is PCoA, not PCA!
3. Why two years that were very different meteorologically, and/or two growing conditions were pooled?
Results: Badly presented. Generally mixed with Discussion.
Discussion: Practically missing.
Conclusions: Why something happened and how?
References: Written differently, not homogenized. In several phrases the verb is missing. Many ideas are pooled in one paragraph. Use present for published papers/chapters and verbs in past for your results.
Detailed observations you can find in the pdf version.

Reviewer 3 Report
The overall introduction to the topic is very good and concise. The experimental part and presentation of results can be improved by correcting some typos (like "CO2 amount"), some synthax aspects (like "the amount of essential oil was converted into the air-dry mass of the herb") and by condensing the text. My main point of concern, however, is the lack of any data about the chemical composition of these basil essential oils whatsoever. I therefore urge that the authors reconsider to rewrite the manuscript by adding some other experimental data as well. I am afraid that with these limited data alone the manuscript will not be of a high scientific interest.